# Investigating the Relation of Intelligence and Executive Functions in Children and Adolescents with and without Intellectual Disabilities

**DOI:** 10.3390/children9060818

**Published:** 2022-06-01

**Authors:** Mieke Johannsen, Nina Krüger

**Affiliations:** 1Educational Psychology and Personality Development, Institute of Psychology, University of Hamburg, 20146 Hamburg, Germany; mieke.johannsen@uni-hamburg.de; 2Department Differential Psychology and Psychological Assessment, Institute of Psychology, University of Hamburg, 20146 Hamburg, Germany

**Keywords:** cognitive abilities, intelligence, executive functioning, structural relation, intellectual disabilities, Intelligence and Development Scales-2

## Abstract

Despite their separate research traditions, intelligence and executive functioning (EF) are both theoretically and empirically closely related to each other. Based on a subsample of 8- to 20-year-olds of the standardization and validation sample (*N* = 1540) of an internationally available instrument assessing both cognitive abilities, this study aimed at investigating a comprehensive structural model of intelligence and EF tasks and at gaining insight into whether this comprehensive model is applicable across sexes and age groups as well as to a subsample of participants with (borderline) intellectual disabilities (IQ ≤ 85, *n* = 255). The results of our exploratory factor analysis indicated one common EF factor that could be sufficiently integrated into the intelligence model within our confirmatory factor analyses. The results suggest that the EF factor can be added into the model as a sixth broad ability. The comprehensive model largely showed measurement invariance across sexes and age groups but did not converge within the subsample of participants with (borderline) intellectual disabilities. The results and implications are discussed in light of the current literature.

## 1. Introduction

Cognitive abilities, their assessment, and their conceptualization are a key interest in psychological [1], neuropathological [2,3], and pedagogical research [4]. While intelligence and its structure have been under investigation for over a century now [5], the concept of executive functioning (EF) was developed more recently, in the 1970s [6]. Due to their different research backgrounds—early intelligence assessment has been linked to aptitude diagnostics [7], while EF research stems from neuropsychopathology [8]—both abilities have only recently been studied within an integrative framework. Nevertheless, the findings suggest that intelligence and EF are closely related to each other [9,10,11,12,13] and relevant within different applicational settings [14,15,16,17,18]. More specifically, research on intellectual disabilities indicates the importance of considering EF when looking at the lower end of the intelligence continuum [18,19,20,21].

Accordingly, this paper aims at (a) investigating the factorial structure of the EF tasks of the IDS-2 [22], (b) testing whether an EF factor could be integrated into the intelligence model of the IDS-2, (c) establishing whether an integrative model of intelligence and EF shows measurement invariance across sex and age, and (d) testing whether the integrative model is applicable to a subsample of children with (borderline) intellectual disabilities.

### 1.1. Intelligence

The long research tradition on intelligence has been intertwined with the development of intelligence measurements ever since [7]. Likewise, defining intelligence is closely linked to the selection of tasks used for its assessment. Following the authors of the IDS-2 [22], we employ the comprehensive definition of Gottfredson [23], which refers to intelligence as a general cognitive capacity including the abilities to reason, to plan, to solve problems, to think abstractly, to understand complex ideas, to learn quickly, and to learn from experience.

Regarding the structure of intelligence, the development of theories is also inevitably linked to the conceptualization and measurement of it [24]. Thus, the research on cognitive abilities and the theory formation advanced along with the development of cognitive tasks and statistical methods. Spearman [5] proposed a general factor of intelligence (g-factor) based on factorial analyses of mental ability tests. However, the question of whether a g-factor exists has been one of the most discussed issues among intelligence researchers [1]. On the one hand, Cattell and Horn [25,26,27,28,29] denied the existence of a single g-factor in their intelligence models including two (fluid and crystallized intelligence) [27] and nine primary cognitive abilities [29], respectively. On the other hand, Carroll [30] included a second-order g-factor and eight subordinate first-order factors in his proposal of a three-stratum model of cognitive abilities attained through data set analysis. McGrew [31] integrated Horn’s extension of the fluid-crystallized model by Cattell and Carroll’s three-stratum model into the Cattell–Horn–Carroll theory of intelligence (CHC theory). The CHC theory therefore provides a recent theoretical framework for intelligence research. It proposes a hierarchical structure of intelligence, including a superordinate g-factor, at least 14 broad abilities at the second stratum, and several subordinate narrow abilities in its latest version [32]. Accordingly, the IDS-2 intelligence domain is hierarchically structured and resembles other intelligence tests such as the Wechsler Intelligence Scale for Children—Fifth Edition [33] and the Stanford–Binet Intelligence Scales—Fifth Edition [34]. However, the IDS-2 [22] intelligence domain includes not only five but seven primary factors which are assessed by 14 subtests and are accompanied by additional modules assessing scholastic aptitudes, EF, and social–emotional skills. The theoretical intelligence model of the IDS-2 is depicted in Figure 1. The model comprises six of the broad abilities of the CHC theory. That is, the IDS-2, in contrast to the CHC theory, differentiates between auditory and visual–spatial working memory components and combines long-term storage and retrieval (corresponding to the former CHC long-term storage and retrieval). Moreover, additional factors of the CHC theory are assessed within the scholastic aptitude module but are not included in this study.

Empirically, the factors structure of the intelligence domain within the IDS-2 seems to differ from the theoretical model [35,36]. Grieder and Grob [35] identified only five primary intelligence factors within the standardization sample of the IDS-2 using exploratory factor analyses (EFA). Their analyses indicated that the verbal reasoning and long-term memory tasks, as well as the visual processing and abstract reasoning tasks, loaded onto common factors that were named Semantic Long-Term Memory and Abstract Visual Reasoning, respectively. However, utilizing the confirmatory factor analyses (CFA) approach, Grieder and colleagues [36] identified a six-factor solution as the best fitting model in their study based on a Dutch and a German sample. This model reinstates the theoretically proposed separation of the Verbal Reasoning factor and the Long-Term Memory factor while allowing for a cross-loading of the Verbal Reasoning factor onto the “Story Recall” subtest. Within the current study, we therefore assume this model, including six broad abilities to represent the factorial structure of the IDS-2 intelligence tasks. 

### 1.2. Executive Functioning

The research on EFs stems from neuropsychological investigations of cognitive functions, especially in patients with lesions in the prefrontal cortex [37,38,39]. EFs are consistently associated with the activity of the frontal brain areas, and this neurostructural association can be seen as a neuropsychological definition of EF [40]. Cognitive researchers claimed a lack of a common functional definition of EF for a long time [37,41]. More recently, there have been attempts to formulate a comprehensive definition [8,40,42]. Accordingly, the term EF functionally refers to a collection of domain general control and regulation mechanisms enabling goal-oriented and situational adjusted behavior and are needed if situations require a deviation from ingrained behavioral patterns like automatic, instinctive, or intuitive responses [8,40,42].

The question whether EF is a unitary construct or if there are more than one separable executive abilities is a central issue of EF research [8,38,43]. Baddeley and Hitch [44] first described the concept of EF as the “central executive” in their multi-component working memory model. Some researchers even equate EFs with working memory processes [8]. Others, however, argue that there are components of EF going beyond working memory processes [38,42]. That is, EF comprises abilities going beyond mere working memory capacity, as they support behavior initiation and inhibition. More specifically, Miyake and colleagues [38] approached the question of the conceptual structure of EF using confirmatory factor analysis (CFA) to investigate the underlying latent variables in an adult sample. They found three core processes of EFs which were “moderately correlated with one another, but […] clearly separable” (p. 49), indicating both the unity and diversity of EFs. While some findings contradict these results [10,12,45], others reinforced the proposed three factor model of EF [8,42,43,46,47]. Diamond [42] endorsed a structure comprising these three core EF factors in her review proposing the following names: (1) inhibition, (2) cognitive flexibility, and (3) working memory. Accordingly, inhibition refers to the ability to deliberately overcome dominant, automatic, inappropriate, or useless inner prerequisites or outer enticements by controlling one’s behavior, thought, attention, or emotions [38,42]. Cognitive flexibility refers to the flexible adjustment to new circumstances and the changing of the view on problems [42]. Working memory refers to the ability to hold information in memory and mentally modify it [42]. It thus goes beyond the mere maintenance of information in short-term memory, as it involves the dynamic manipulation of working memory content [38]. Additionally, Diamond [42] differentiated between these three basic EFs and higher order EFs such as reasoning abilities that can be equated with fluid intelligence, one of the broad abilities within the CHC model. 

EFs, in general, increase from preschool over school age until early adulthood [39,45]. Research suggest a differential development of EFs during childhood and adolescence, however [37,41,43,45]. While Diamond [42] states that cognitive flexibility develops in later childhood only, Huizinga and colleagues [45] identified a two-factor model including cognitive flexibility and working memory but no inhibition factor in children aged eight years or older, indicating a structural stability from that age onwards. Accordingly, this work includes only data of children that are 8 years or older. 

Within the IDS-2 [22], EF is assessed within four subtests that tap different aspects of the described core EF factors. The first subtest—“Listing Words”—is a verbal fluency test that is meant to assess cognitive flexibility. The second subtest—“Divided Attention”—combines a verbal fluency task with a processing speed task and hence requires abilities that relate to working memory and cognitive flexibility. The third subtest—“Animal Colors”—is a variation of the Stroop-test, a well-known interference task. Accordingly, it measures the ability of inhibition. The fourth and last subtest—“Drawing Routes”—is a planning task that requires working memory capacity as well as inhibition. As these tasks were not intentionally developed to assess the three core EF factors separately, the factorial structure of the EF domain within the IDS-2 will be addressed within this study. Due to the conceptual overlap of the tasks, they do require different aspects of EF simultaneously, and a one-factor structure is assumed.

### 1.3. Relation of Intelligence and Executive Functions

There are heterogeneous findings on the conceptual overlaps and disparities between intelligence and EFs. Despite their separate research backgrounds, intelligence and EFs are conceptually closely linked to each other. Many researchers state that the definitions of both constructs coincide [8,10,13,42,48]. The definitions employed in the present work partly resemble each other in that both refer to general cognitive mechanisms involving the ability to learn from (intelligence) or adapt to experiences (EF). Moreover, goal-oriented behavior (EF) requires planning and problem solving (intelligence). Definitional overlaps like this can be found a lot when looking at different definitions of both constructs and illustrate their conceptual relatedness.

Moreover, the tasks used to asses EFs are mostly not only tapping one specific EF but rather involve different cognitive abilities including other executive as well as nonexecutive processes such as intelligence [8,40,41,43,46]. Therefore, the correlational analysis of the relations between EFs and intelligence shows rather inconsistent results [9]. The findings concerning the three core EFs point towards a differential relationship [13,48]. Friedman and colleagues [48] (2006) found that working memory was most closely related to intelligence compared to inhibition and cognitive flexibility. The tasks used to assess EF and fluid intelligence require the manipulation of information in working memory. This indicates a crucial role of working memory in the relation of EF and fluid intelligence, however. 

In addition to the differentiation between the different EFs, research suggests a differential relation of fluid and crystallized intelligence to EFs as well. Some researchers found that EFs are more closely linked to fluid than to crystallized intelligence [49,50]. Others did not find such a differential relationship but rather argued that a latent EF variable accounts for the commonality of fluid and crystallized intelligence [10,48]. These results might indicate the resemblance of EF in their relationship to fluid and crystallized intelligence either to other first-order intelligence factors or to a common g-factor. However, there are no studies investigating a common structure of intelligence and EF including several primary intelligence factors yet. Thus, this study aims at investing the common factorial structure of the intelligence and EF domains within the IDS-2.

Besides the mentioned conceptual and functional relations of intelligence and EF, they show genetical, neurobiological, and developmental associations too. Intelligence and EFs are both highly heritable [12,46,51]. Engelhardt and colleagues [12] recently found that the genetic influences on intelligence overlap with those on EF. EF was strongly associated with intelligence in a genetically mediated way, even after controlling for processing speed. Additionally, EFs and intelligence partly share a neurobiological basis as well. While both are associated with neural activity in a shared frontoparietal network, they do have individual neural correlates as well [11]. This shared genetical and neurobiological basis may be reflected in a g-factor that influences both EF and primary mental abilities subsumed as intelligence. Developmentally, research indicates that EF resources predict the later development of reasoning skills [52,53]. While giving no clear insight on the development of the structural relation, these results indicate that, on the one hand, EF is reasonably distinguishable from intelligence, and, on the other hand, EF appears to be an antecedent of (fluid) intelligence development. Accordingly, we assume that the EF domain can be integrated as an additional broad ability into the IDS-2 intelligence model and will test the resulting model for measurement invariance across age to account for possible changes in the structural relationship of intelligence and EF.

### 1.4. Intellectual Disabilities

The concept of intellectual disabilities is by its very nature linked to the concept of intelligence. Definitions provided by the World Health Organization [54] and the Diagnostic and Statistical Manual of Mental Disorders [55] characterize an intellectual disability or intellectual developmental disorder as a significant impairment in general cognitive functioning, social skills, and adaptive behavior. This “significant impairment is characterized as performance that is [two] or more standard deviations below the mean based on normed, individually administered standardized tests of cognitive and adaptive function” [56] (p. 2). Therefore, an individually administrated, standardized psychometric test, such as the IDS-2, resulting in an intelligence quotient (IQ) of 70 or under is a main indicator of intellectual disabilities and is commonly used for the operationalization of intellectual disabilities in research [57,58]. Complete clinical diagnostics require additional information on social skills and adaptive behavior, however.

Research indicates that the cognitive functioning in intellectual disabilities is not only characterized by a very low level of overall intelligence but also by a differential structural relation among cognitive abilities [57,58] and impairments of specific cognitive functions [59]. More specifically, individuals with intellectual disabilities also show deficits regarding their EF [18,20]. Meta-analytically, Spaniol and Danielson [18] found that people with intellectual disabilities showed statistically significantly lower EF in comparison to mental age-matched control groups. Moreover, the work of Schuchhardt and colleagues [20] indicates that specific deficits in EF are already present in children with borderline intellectual functioning characterized by an IQ above 70 but below 85. These results indicate that EF plays a vital role when looking into the cognitive functions associated with lower levels of cognitive functioning that are characteristic of (borderline) intellectual disabilities. Thus, the current study aimed at investigating whether the relevance of EF for intellectual disabilities including borderline intellectual functioning is reflected in a different structural relationship of intelligence and EF in individuals with a full-scale intelligence quotient equal to or below 85.

### 1.5. Research Aims

Based on the theoretical considerations mentioned above, this study targeted the following four research aims.

We aimed at investigating the factorial structure of the EF tasks of the IDS-2. Due to the conceptual overlap of the tasks requiring different aspects of EF simultaneously, a one-factor structure was assumed. However, this research question will be addressed using EFA, as there are no finding on the factorial structure yet.We investigated whether the identified EF factor(s) could be integrated into the empirical model of the IDS-2 intelligence tasks proposed by Grieder and colleagues [36]. Based on past research, EF factors are closely linked to intelligence, and it can be assumed that a common EF factor might be comparable to other primary factors of intelligence. To test this assumption, we used a common second-order CFA of the intelligence and EF tasks and hypothesized that a model with a freely estimated relation of the second-order g-factor to the first-order EF factor fits the empirical data best compared to a model with a fixed relation (equal or unrelated).We tested an integrative model for measurement invariance regarding sex and age. Measurement invariance testing allows for the determination of whether a construct assessed by an psychometric test is measured equivalently across groups [60]. Since the new model including both intelligence and EF has not yet been tested for measurement invariance, we aimed at investigating whether the model showed measurement invariance across age and sex and is thus applicable to these groups.We investigate whether the structural relations of EF and intelligence differ in children with (borderline) intellectual disabilities. As the research indicates, children with (borderline) intellectual disabilities show specific deficits in EF, and it is therefore of interest whether the identified structural relationship differs within a subsample of children with (borderline) intellectual disabilities in comparison to the rest of the intelligence continuum. To test this hypothesis, the fitted integrative model will be tested for measurement invariance across a subsample of children with (borderline) intellectual disabilities.

## 2. Materials and Methods

### 2.1. Participants

The current work is based on the sample (*N* = 2030) of the standardization and validation of the IDS-2 [22] that took place from 2015 until 2017 in Switzerland, Germany, and Austria. Due to the structural stability of EF being reached at the age of 8 years (according to previous research findings [45]), only the data of participants aged 8 years or older were included. Therefore, the total sample for our analyses included 1540 participants (*M*_age_ = 13.79, *SD*_age_ = 3.78, 52.1% female) Moreover, a subsample including only participants with an IQ of 85 or below (*n* = 255, *M*_age_ = 13.73, *SD*_age_ = 3.66, 43.9% female) was used to test our hypothesis regarding the measurement invariance of the model across (borderline) intellectual disabilities. The sample division was primarily based on the complete IQ profile, including all 14 subtests. In case the complete IQ profile was not available due to missing values in some subtests (*n* = 22), the IQ score calculated on the basis of the first seven subtests was used to avoid dropouts. This score provides a sufficient assessment of the overall cognitive abilities as well [22]. While this subsample did not differ regarding its mean age (*t* (372.6) = 0.24, *p* = 0.814, *d* = 0.02), there were fewer girls in the subsample (*t* (365.75) = 2.87, *p* = 0.004, *d* = 0.20). Moreover, chi-square tests for residency (χ² (2) = 43.89, *p* < 0.001, *Cramer’s V* = 0.17) and maternal educational status (χ² (5) = 200.33, *p* < 0.001, *Cramer’s V* = 0.37) indicated medium and large differences between the samples regarding these demographics, respectively. For further information on the demographics of the full sample and the subsample, see Appendix A.

### 2.2. Materials and Procedure

The data collection for the standardization and validation of the IDS-2 [22] took place in Switzerland, Germany, and Austria. Schools and psychosocial institutions for children and adolescents carried out the participant recruitment. The administration of the whole test battery took between 3.5 and 4.5 h and was split into two sessions no more than 1 week apart, if necessary. The parents (for 8- to 15-year-olds), or both the participants and their parents (for 10- to 20-year-olds), gave written consent for participation. Moreover, the adolescents or parents provided demographic information within a personally administered questionnaire at the beginning of the data collection. In exchange for their participation, the participants received either a gift card (Switzerland; CHF 30) or EUR 25 in cash (Germany and Austria). The Ethics Committee Northwest and Central Switzerland, as well as the responsible local ethics committees in Switzerland, Germany, and Austria, granted ethical approval for the data collection.

The IDS-2 [22] is a modularly constructed test battery developed for the assessment of cognitive abilities (intelligence, EF) and further developmental functions (psychomotor skills, socioemotional competence, scholastic skills, and attitude toward work) in 5- to 20-year-olds. The test battery comprises 30 subtests in total, 14 of which are used for measuring the seven intelligence factors and an additional 4 assessing the EF. Brief descriptions of the intelligence and EF subtests are given below.

#### 2.2.1. Intelligence

The theoretical IDS-2 intelligence model includes seven factors reproducing six of the CHC broad abilities. The factors are assessed by two subtests each, resulting in a total of fourteen intelligence subtests. The Visual Processing factor refers to the ability to perceive, analyze, store, and retrieve visual stimuli. It is assessed with the subtest “Geometric Shapes”, requiring the reproduction of presented figures using given rectangles or triangles, and the subtest “Plates”, requiring the reproduction of a figure made up of round platelets on a magnetic pad. 

Processing Speed refers to the automaticity and fluency of executing cognitive tasks. Within the subtest “Two Characteristics”, the participants must cross out parrots with two orange body parts looking towards the left in a limited time per row. “Crossing Out Boxes” requires the participants to mark groups of shapes containing exactly four squares. 

Short-Term Memory is divided into two factors in the IDS-2: Auditory Short-Term Memory and Visuo–Spatial Short-Term Memory. Both refer to the ability to process, store, and retrieve information in and from short-term memory, the former referring to auditory information and the latter referring to visuo–spatial information. Auditory Short-Term Memory includes the subtests “Numbers/Letters” and “Numbers/Letters Mixed”. Within both subtests, the participants repeat series of numbers or letters (alternately or mixed, respectively), first in chronological and then in reverse order. The subtests used to assess Visuo–Spatial Short-Term Memory are “Geometric Figures” and “Rotated Geometric Figures”. In both subtests, the participants must remember figures presented to them and select these target figures out of several different figures. In the second subtest, the figures are rotated between presentation and recognition.

Abstract Reasoning refers to logical thinking and reasoning ability. In the subtest “Completing Matrices”, the participants must logically infer a missing component in a matrix and choose the right solution out of five possibilities. In the subtest “Excluding Pictures” (EP), the participants must pick one out of six figures that does not fit the others. 

Verbal Reasoning refers to the acquisition and application of knowledge. It is assessed by the subtests “Naming Categories” and “Naming Opposites”. In the former, the participants name the generic concept of three words. In the latter, the participants must name the opposite to a term used in an example sentence. 

Long-Term Memory refers to the storage and retrieval of verbal and visual information in and from long-term memory. It is assessed by “Retelling a Story” and “Describing a picture”. The participants are asked to remember a story read out or a picture shown to them, respectively, and recall it after a period of time. 

Within our sample, all of the intelligence subtests (Cronbach’s alpha: 0.81–0.93), as well as the IQ profile (Cronbach’s alpha = 0.89), showed good to excellent internal consistencies. Moreover, the validity of the IQ scales is well supported [22].

#### 2.2.2. Executive Functioning

There are four tasks assessing EF in the IDS-2. The subtest “Listing Words” is a verbal fluency test requiring the participants to name as many words as possible referring to a given topic or starting with the same letter. The number of correct words is counted. According to the test manual, this task measures cognitive flexibility.

The subtest “Divided Attention” is a combination of the previously described task “Listing Words” and the Processing Speed task “Two Characteristics”. The participants must cross out parrots with the above-mentioned characteristics while naming words fitting given criteria. This task requires abilities that relate to working memory and cognitive flexibility.

The subtest “Animal Color” is a variation of the Stroop test and is thus an interference task. The participants must name the color of animals presented to them. In the first trial, the presented animals are colored in their original color. In the second trial, the presented animals are colored grey, and the participants must name the original color of the animals. In the third trial, the animals are colored in incoherent colors and the participants are still asked to name the original color. Accordingly, this task measures the ability of inhibition.

The subtest “Drawing Routes” is a planning task. The participants must trace the way through a line network. Each line must be traced exactly one time. The raw scores of all four of the subtests are transformed into standardized value points. The composite EF score is the mean of the value points of the four EF tasks.

Regarding the EF measures, Cronbach’s alpha is no sufficient estimate of the reliability for most subtests. Therefore, the reliability of these tasks was not specifically assessed within the current study. However, the test manual [22] provides retest reliability coefficients for all four subtests which indicate good reliability. Moreover, the reported correlations with the other EF measures indicate convergent validity.

### 2.3. Statistical Analyses

The data analyses for this work were conducted using the open source software R [61]. We used the psych package [62] for descriptive analyses, reliability calculations, and EFA, the lavaan package [63] for all CFAs, the semTools package [64] for model comparisons, and the semPlot package [65] for CFA plots.

Due to differences in the metric of the raw scores of the subtests, we used the standardized value points (value points are normally distributed with a mean of 10 and a standard deviation of 3) for all of the statistical analyses of this work. The value points were z-standardized to ensure the better comparability of the estimated parameters. Due to the use of standardized values, the requirement of the metric scalation of the data for CFA [66] is assumed. The EF subtests “Divided Attention” and “Drawing Routes” both include two different scores. Thus, we used the mean of the value points of these two scores as the total subtest score. 

As the tests for normality were assumed to be significant due to the sample size, the univariate and multivariate normality of all the indicators was assessed by examining the skewness and kurtosis and by visually examining the distribution of each variable. As a rule of thumb, a maximum likelihood (ML) estimation was considered adequate for data with an absolute value < 2 for skewness and an absolute value < 7 for kurtosis [67]. The examination of the univariate skewness and kurtosis did not reveal any deviations from normality. However, Mardia’s multivariate skewness and kurtosis estimates indicated multivariate non-normality. The multivariate outliers were examined using the Mahalanobis distance [68]. The exclusion of the multivariate outliers reduced kurtosis, but the test for skewness remained significant. The multivariate outliers (*n* = 85) were kept within the sample, though, as there was no reason to question the validity of the data. Despite the multivariate non-normality, the maximum likelihood estimation was used. The lavaan package provides robust maximum likelihood estimators (“MLR”) suitable for non-normal data [63]. Moreover, the missing data were handled using the full information maximum likelihood (FIML) method in order to use all the information available in the data [69]. 

#### 2.3.1. Measurement Models

We utilized a structural equation modelling (SEM) approach to test our hypotheses regarding the structure of intelligence and EF. To do so, we first examined the measurement models for both constructs separately. Intelligence and EF are conceptualized as reflective measurement models following a factor analytic approach, proposing the latent variables as having a causal influence on their associated manifest indicators [68]. Regarding intelligence, the psychometric property of the proposed model has already been evaluated by Grieder and colleagues [36]. Therefore, the six-factor model identified by them served as the measurement model for intelligence, and its fit within our sample was reassessed utilizing a confirmatory factor analysis approach.

Regarding EF, however, there are no studies on the factorial structure available yet. Thus, we initially utilized exploratory factor analysis to determine the factorial structure. Firstly, the factorability was assessed using the Bartlett test, the measures of sampling adequacy (MSAs), and the Kaiser–Meyer–Olkin criterion (KMO [68]). Afterwards, we determined the number of factors to be extracted based on multiple criteria. The criteria used were (a) Kaiser’s criterion (eigenvalues > 1), (b) Cattell’s scree test, and (c) Horn’s parallel analysis. Subsequently, the principal axis analysis was used to examine the covariance matrix of the EF tasks, and the retained factor was subjected to promax rotation for an oblique factor solution. Finally, the model fit of the identified factor solution was evaluated using CFA.

#### 2.3.2. Structural Equation Models

After the evaluation of the measurement models, we tested our hypotheses using an SEM approach. Firstly, we estimated a common first-order CFA (M1) including both the intelligence and the EF factor(s). Secondly, we calculated a second-order model including a superordinate g-factor loading onto all the primary factors (M2). This second-order model was tested against two nested models with fixed relations of the g-factor to the common EF to investigate whether the EF factor could be reasonable integrated as another broad ability. In the first nested model (M2a), the loading of the g-factor on the EF factor was fixed to zero. In the second model (M2b), the loading of the g-factor on the EF factor was fixed to one. Illustrations of all the tested models can be found in Appendix A.

To ensure model identification, the variances of the latent variables were fixed to unity so that all the factor loadings were estimated freely within the CFAs [63]. All other parameters not explicitly fixed for model comparisons remained free and were estimated within the analysis.

#### 2.3.3. Invariance Testing

The possible moderating influences of sex and age were examined using measurement invariance testing. The measurement invariance testing allows for the determination of whether a construct assessed by a psychometric test is measured equivalently across groups [60]. Since a comprehensive model including both intelligence and EF has not yet been tested for measurement invariance, the best fitting model was tested for measurement invariance across sex and four age groups: 8–10, 11–13, 14–16, and 17–20 years. Moreover, we tested our last hypothesis by examining whether measurement invariance could be established across a subsample of participants with (borderline) intellectual disabilities.

We evaluated measurement invariance based on multiple-group CFAs in accordance with Putnick and Bornstein [60]. Therefore, we fitted a series of models with an increasing number of parameters that were constrained to be equal across groups. That is, we consecutively tested the model for configural invariance (equivalence of the model form), metric invariance (equivalence of factor loadings), scalar invariance (equivalence of intercepts), and strict invariance (equivalence of residual variances).

#### 2.3.4. Fit Indices and Model Comparison

The model fit was evaluated by employing the following fit indices and their cut-off values suggested by Hu and Bentler [70]: root mean square error of approximation (RMSEA), standardized root mean square residual (SRMR), and comparative fit index (CFI), A value of RMSEA ≤ 0.06, a value of SRMR ≤ 0.05, and a value of CFI ≥ 0.95 indicate a good overall fit. For all of the fit indices, the robust variants provided by the lavaan package [63] were used for the model evaluation. The chi-square test, though reported in the result section, was not interpreted due to its sensitivity to large sample sizes [71]. Additionally, the Akaike information criterion (AIC) and the Bayesian information criterion (BIC) were used for the model comparisons [68]. Measurement invariance was assumed when the differences between the fit criteria lay beneath the following cutoffs suggested by Chen [72]: ΔCFI < −0.01, ΔRMSEA < 0.015, and ΔSRMR < 0.01. In the case of non-invariance, we subsequently released single parameters to establish partial invariance.

## 3. Results

### 3.1. Descriptive Statistics

Descriptive statistics of the manifest indicators are depicted in Appendix A.

### 3.2. Measurement Models

#### 3.2.1. Intelligence

The six-factor model suggested by Grieder and colleagues [36] showed a good fit in our sample (χ² (61) = 110.51, *p* < 0.001, RMSEA = 0.023, SRMR = 0.014, CFI = 0.994). An illustration of this model, including parameter estimates, can be found in Appendix A. 

#### 3.2.2. Executive Functioning (EF)

The significance of the Bartlett test (χ² (6) = 1414.63, *p* < 0.001), the MSAs ranging from 0.62 (AC) to 0.79 (DA) and thus being above the proposed cut-off value of 0.5 [68], and the KMO of 0.67 (cut-off value: 0.6) indicated the suitability of the data for EFA. All of the EFA criteria used indicated a one-factor solution to be the most suitable. The subsequent principal axis analysis subjected to promax rotation yielded factor loadings ranging from of 0.30 (PR) to 0.91 (AC). For an overview of all the factor loadings, see Appendix A. The CFA of the resulting one-factor EF model yielded a good fit (χ²(2) = 5.65, *p* = 0.059, RMSEA = 0.035, SRMR = 0.012, CFI = 0.997). Therefore, the assumed one-factor structure of the IDS-2 EF task was clearly supported by the data.

### 3.3. Structural Equation Modeling

The fit measures for the models testing the relation of intelligence and EF are depicted in Table 1. In the first step, the first-order CFA (M1), including the six intelligence factors and the EF factor, was estimated. Secondly, the second-order model (M2), including a superordinate g-factor, was calculated. The fit measures indicate that the model M2 fitted the data well. In the third step, the loading of the g-factor onto the EF factor within the second-order model was fixed to 0 (M2a) and 1 (M2b), respectively. While the fit indices of M2a indicate an insufficient fit and thus an inappropriateness of the model, those of M2b yielded a marginally good fit, with the RMSEA and SRMR being slightly above the suggested cut off. Finally, the model comparison indicated that, out of M2, M2a, and M2b, the second-order CFA with a freely estimated loading of the g-factor onto the EF factor (M2) showed the best fit according to all the fit measures.

### 3.4. Invariance Testing

As model M2, with a freely estimated loading of the g-factor onto the EF factor, showed the best fit among the second-order models, it was subsequently tested for invariance across sex and age as well as within the (borderline) intellectual disability subsample. Table 2 shows the results of the invariance testing of M2.

Regarding sex, the configural and metric invariance of M2 could be established. However, constraining the intercepts to be equal across the groups led to a relevant decrease in the model fit, indicating that the model showed no scalar invariance. Partial invariance was acquired by releasing the equality constraint for the intercept of the subtest “Geometric Shapes”. Including this alteration, the model subsequently reached strict invariance. This finding suggests that the overall factorial structure, the factor loadings, the intercepts except the one of “Geometric Shapes”, and the residual variances can be assumed to be equal for boys and girls. 

Regarding age, configural measurement invariance was established. However, constraining the loadings to be equal across groups led to a relevant decrease in the model fit, indicating that the model showed no metric invariance. Partial metric invariance was acquired by releasing the equality constraint for the loading of the Visuo–Spatial Short-Term Memory factor onto the “Rotated Geometric Figures” subtest. This was reflected in an increase in this loading with increasing age. Including the alteration, the model subsequently reached strict invariance.

Configural invariance of the best fitting model across ability groups could not be established due to determination problems (e.g., Heywood case). Thus, the metric, scalar, and strict invariance of this model could not be evaluated. As the determination problems might reasonably have been caused by the cross-loading of the Verbal Reasoning factor onto the “Retelling a Story” subtest [36], we ran exploratory analyses on the invariance of the model without this cross-loading. The invariance testing of this model was possible without determination problems. Nevertheless, configural invariance could not be established (χ² (256) = 721.26, *p* < 0.001, RMSEA = 0.049, SRMR = 0.047, CFI = 0.918). To gain further insight into the factorial structure of the intelligence and EF tasks within the (borderline) intellectual disability subsample, we additionally repeated the model fitting steps reported above within this subsample. As Mardia’s test indicated multivariate non-normality, the multivariate outliers were excluded for the subsequent analyses, resulting in a reduced sample (*n* = 220). While the intelligence measurement model including six broad abilities [36] but no cross-loading resulted in Heywood cases within the subsample, the measurement model including only five broad abilities [35] fit the data well (χ² (67) = 98.58, *p* = 0.007, RMSEA = 0.045, SRMR = 0.037, CFI = 0.974). The EFA on the EF subtests yielded one common factor. Subsequent analyses regarding the integration of the EF factor into the intelligence model resulted in findings similar to those of our main analyses, with a model freely estimating the loading of the g-factor onto the EF factor fitting the data best. For a more detailed overview of these exploratory results, see Appendix A.

## 4. Discussion

The aim of this study was to (a) investigate the factorial structure of the EF tasks of the IDS-2 [22], (b) test whether an EF factor could be integrated into the intelligence model of the IDS-2, (c) establish whether an integrative model of intelligence and EF shows measurement invariance across sex and age, and (d) test whether the integrative model is applicable to a subsample of children with (borderline) intellectual disabilities.

Our results indicate that (a) there is one common EF factor explaining the different amounts of variance of each EF task and that (b) this common factor can be integrated into the intelligence model identified by Grieder and colleagues [36] as another broad ability. Moreover, (c) the integrative model was largely invariant across sex and age groups but (d) did not converge within the (borderline) intellectual disability subsample. 

### 4.1. Factorial Structure of EF

All of the criteria used within our EFA indicated a one-factor solution regarding the factor structure of the EF tasks. This clearly indicates that the EF tasks within the IDS-2 commonly require certain EF abilities. However, our results do not give clear insight into which core EF the common EF factor might represent within the nomenclature of Diamond [42], as the IDS-2 EF tasks do simultaneously tap different EF abilities. On the one hand, the identified factor might reasonably reflect a superordinate general EF ability that is relevant to all the tasks [10,12,45]. On the other hand, based on the yielded factor loadings, we argue that the common EF factor could also reasonably represent an inhibition ability, as it showed the highest loading onto the “Stroop-like Animal Colors” subtest. Nevertheless, it might also reflect cognitive flexibility, as it also showed moderate loadings onto both the “Listing Words” and “Divided Attention” subtests. Therefore, future studies should more precisely disentangle the abilities assessed within the IDS-2 EF tasks by including additional EF tasks.

### 4.2. Structural Relation of Intelligence and EF

Based on the close relation of intelligence and EF reported in previous research [8,40,41,43,46], we assumed that a latent EF factor might be reasonably integrated into the intelligence model of the IDS-2. The model comparisons of three second-order CFAs varying in their constraints of the loading of the g-factor onto a common EF factor revealed that a model with a freely estimated g-loading onto the EF factor fit the data best. That is, both a fixation of the loading to 0 or 1, respectively, led to a decrease in the model fit, indicating that the relation of the g-factor to the EF factor is comparable to the association of the g-factor to the other intelligence factor. The considerable decrease in fit when the loading was fixed to zero especially indicates that the EF factor is related to a second-order intelligence factor. 

This finding is in line with neurobiological and genetical research on the intelligence–EF relation [12], which indicates a common genetical basis of intelligence and EF that structurally might be represented by the g-factor. Interestingly, the estimated loadings of M2 indicated that the g-factor was most strongly associated with the EF factor compared to the other broad abilities. This underlines both the theoretical closeness of general intelligence and EF and the relevance of EF for the practical assessment of general cognitive abilities.

Based on these close relations, one may reasonably assume that EF can be integrated into the CHC theory as another broad ability factor. However, we argue, in line with Schneider and McGrew [32], that EF, rather than being an additional broad ability, is closely related to different broad intelligence factors. That is, EF abilities are required to successfully complete the different tasks associated with the different broad abilities. For example, executive processes related to the manipulation of information (working memory) are a prerequisite for both short-term and long-term memory. 

Moreover, one may be concerned with the necessity to separately assess EF, given that it is so closely related to (general) intelligence. However, our results also indicate that, while closely related, EF is still different from (general) intelligence. In addition, the variability in the factor-loadings of the EF tasks indicates that they differ in the respective EF abilities required. Therefore, given the specific importance of EF for not only intellectual disabilities [18,20] but also everyday functioning [42], the practical relevance of specific EF measures for different applicational settings is undeniable. The EF subtests included within the IDS-2 [22] enable practitioners to sufficiently assess different aspects of EF within a module that can be administered separately from the intelligence tasks.

### 4.3. Invariance across Sex and Age

Regarding sex, full metric invariance was supported. This finding indicated that the same constructs are measured across sexes. In line with previous studies on the German IDS-2 [22,36], only partial scalar invariance was supported by our analyses. This finding might be explained by the sex differences in visual–spatial abilities, which indicate an advantage of males compared to females [73,74]. Based on the partial invariance of intercepts, strict invariance was also supported, indicating that the residual variances are comparable across groups and therefore that the constructs’ reliabilities are also comparable. Overall, the establishment of measurement invariance for an integrative intelligence–EF model (except for one intercept) implies the applicability of the intelligence and EF tasks and their structural relation to both sexes and the comparability of the norms for males and females.

Regarding age, configural invariance was supported. However, the test for metric invariance and the subsequent establishment of partial metric invariance indicate that the loading of the Visual Short-Term Memory Factor onto the “Rotated Geometric Figures” subtest differed between the groups. This loading was lower in the youngest group, thereby suggesting that the “Rotated Geometric Figures” subtest might be less indicative of Visual Short-Term Memory in younger children. While the age invariance testing reported in the test manual [22] did not indicate any age differences, Grieder and colleagues [36] found a similar result. A possible reason for this might be the ongoing development of visual working memory from childhood to adolescence [75]. The “Rotated Geometric Figures” subtest requires the mental rotation of the visual stimuli and therefore a more pronounced ability than the “Geometric Figures” subtest, where no rotation of the stimuli is necessary. Therefore, these two subtests assessing Visual Short-Term Memory might share less variance in younger children, whose mental rotations are not as developed as those of older children [76,77].

### 4.4. Applicability to (Borderline) Intellectual Disability

Configural invariance of the integrative model across a subsample of (borderline) intellectual disability could not be established due to determination problems. These problems might reasonably be caused by the cross-loading of the Verbal Reasoning factor onto the “Retelling a Story subtest”, as suggested by Grieder and colleagues [36]. We therefore ran exploratory analyses on the invariance of the model without this cross-loading and repeated the model fitting process within this specific subsample. The results indicated that the model that fit best within the complete sample could not be sufficiently estimated within the (borderline) intellectual disability subsample. Interestingly, the structure of the intelligence domain seems to be slightly different within the (borderline) intellectual disability sample, as a model including only five broad abilities [35] fit the data well, while the assumed six factor model resulted in Heywood cases. However, these results should be interpreted cautiously, as these analyses were conducted exploratorily. Nevertheless, these results indicate that there may actually be specificities in the factorial structure of the cognitive abilities assessed within the IDS-2 [22] in a (borderline) intellectual disability sample. This finding is in line with previous research on (borderline) intellectually disabled samples, which indicated a less pronounced differentiation of the primary mental abilities associated with intellectual disability [57,58]. Still, the limitations regarding this part of our study (described below) should be considered and addressed in future research. 

### 4.5. Strength and Limitations

There are certain limitations to this study. First, one could argue that the investigation of a comprehensive structural model of intelligence and EF should include a range of different tests to assess these abilities rather than a single test battery. While the representativeness of the IDS-2 [22] model regarding the CHC theory is certainly limited with regard to the number of broad abilities and the number of indicators per factor, the data basis of this study can also be considered its greatest strength. The use of the standardization and validation sample of the IDS-2 enabled us to investigate a comprehensive structural model of intelligence and EF within a representative sample covering a considerate age range. Therefore, the results can be assumed to be applicable to the population of 8- to 20-year-olds across Germany, Austria, and Switzerland. Moreover, the use of a common instrument that assesses both constructs facilitates the transfer to practical applications (see Section 4.6).

Furthermore, the following limitations apply to the investigation of our fourth research question. First, the multiple-group CFA resulted in Heywood cases, and invariance could therefore not be evaluated. This determination problem might be explained by the division into groups being based on the full-scale IQ. This index itself is based on the indicators used within our model, and the subsample was therefore restricted in its variance of these indicators. Nevertheless, the determination problems were ameliorated by dropping the cross-loading within the model, and this modified model still did not show configural invariance. Moreover, additional exploratory analyses further supported the assumption of differential structural relations among cognitive abilities within a sample of (borderline) intellectual disability. Second, the size and representativeness of the subsample was restricted. The limited sample size (*n* = 255) might have led to a restriction of power. As the data were not exclusively collected for this study and only data available within the standardization sample of the IDS-2 [22] were used, a priori power analyses were not applicable. Still, the sample was larger than that of comparable studies [57,58], and our explanatory analyses yielded interpretable results underlining the sufficiency of the sample size. Our results further indicated that the subsample differed with regard to maternal education and residency; therefore, there might be confounding effects of these socio-demographic variables. Parental education has especially been shown to be associated with differences in intelligence [78,79]. This issue might be addressed by future research comparing the structural relation between an intellectual disability and a matched control group. Third, the operationalization of (borderline) intellectual disability in the current study does not fully correspond with the diagnostic criteria [54,55], as it solely included an indicator of cognitive functioning and included no indicators of social skills or adaptive behavior. Thus, one might argue that the results are not completely applicable to children with intellectual disabilities. However, we would like to argue that cognitive functioning is nevertheless a key element of the diagnostic criteria of intellectual disabilities, and, therefore, our results contribute to a better understanding of intellectual disabilities and should also be considered in practical applications (see Section 4.6). Still, future research on the sufficiency of the IDS-2 assessment in intellectual disability should utilize a more comprehensive operationalization based on complete clinical assessments. 

### 4.6. Implications

Our results have implications for theory, future research, and practice. Theoretically, our results strongly support the structural relatedness of intelligence and EF, especially of a superordinate g-factor and an EF factor. Considering this relatedness, it seems reasonable to assess EF in application settings when general cognitive abilities are relevant. The IDS-2 provides the opportunity to assess both intelligence and EF within one test battery. Therefore, the IDS-2 is advantageous in comparison to separate assessments because it provides the sufficient comparability of both abilities, as the standardization of both domains is based on the same sample. 

Moreover, results of the invariance testing indicated that the comprehensive model is largely invariant across sex and age groups, and the model thereby provides scores of cognitive abilities that are comparable across sexes and ages. Therefore, the comprehensive assessment of intelligence and EF within the IDS-2 is suitable for a wide range of practical applications and thereby fulfills the need for a practically applicable instrument that is indicated by the relevance of EF in developmental disorders, everyday functionality, and school and life success [42].

However, the unity of the EF factors indicates that the EF tasks used within the IDS-2 are quite homogenous and are therefore not representative of all the core EFs identified by past research [38,42]. Therefore, future studies should more precisely disentangle the abilities assessed within the IDS-2 EF tasks. Moreover, additional information on specific EF abilities may be useful when assessing the specific deficits of EF associated with intellectual disabilities [20,21], both in practical settings and in future research. Additionally, future research needs to replicate the current findings while addressing the above-mentioned limitations and thereby ensuring the validity and timeliness of the findings. 

## 5. Conclusions

The current study aimed at investigating the common factorial structure of intelligence and EF as they are assessed within the IDS-2 [22]. The results support a common factorial structure of these abilities that is widely invariant across age and sex. However, the factorial structure of the assessed cognitive abilities appears to be slightly different in a subsample with (borderline) intellectual disability. This finding points towards the need for further research on cognitive abilities and their structure in (borderline) intellectual disabilities and indicates that broad cognitive abilities (assessed within the IDS-2 [22]) should be interpreted cautiously at the lower end of the intelligence continuum. Nevertheless, the IDS-2 provides practitioners with a comprehensive instrument that can be used to assess not only intelligence but also a wider range of cognitive and developmental functions.

## Figures and Tables

**Figure 1 children-09-00818-f001:**
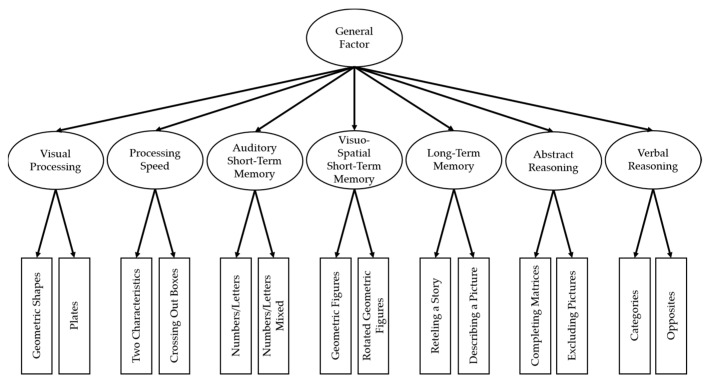
Theoretical Structure of the intelligence domain of the Intelligence and Development Scales-2 [22] with 14 subtests on Stratum I, 7 factors on Stratum II, and a general factor on Stratum III. Figure adapted with permission from Grieder and Grob [35].

**Table 1 children-09-00818-t001:** Maximum likelihood estimation and model fit statistics.

Model	Fit Indices							
	χ²	df	CFI	RMSEA	CI	SRMR	AIC	BIC
M1	363.55	113	0.977	0.038	[0.034, 0.043]	0.027	66,925.86	67,331.61
M2	545.789	127	0.961	0.047	[0.043, 0.051]	0.035	67,083.66	67,414.67
M2a	1587.118	128	0.865	0.087	[0.083, 0.091]	0.187	68,142.45	68,468.12
M2b	631.912	128	0.953	0.051	[0.047, 0.055]	0.065	67,171.59	67,497.26

Note: Variances of the latent variables were constrained to unity to ensure model identification. M1 = first-order CFA including six intelligence factors and one EF factor, M2 = second-order CFA: M1 additionally including the superordinate g-factor, M2a = M2 with the loading of g onto the EF factor fixed to 0, M2b = M2 with the loading of g onto the EF factor fixed to 1. CFI = comparative fit index, RMSEA = root mean square error of approximation, CI = 90% confidence interval for RMSEA, SRMR = standardized root mean square residual, AIC = Akaike’s information criterion, BIC = Bayesian information criterion.

**Table 2 children-09-00818-t002:** Model fit estimates of the invariance testing.

Grouping Variable	Invariance Level	df	CFI	RMSEA	SRMR	ΔCFI	ΔRSMEA	ΔSRMR
Sex	configural	254	0.963	0.046	0.036			
	metric	272	0.962	0.045	0.042	−0.001	−0.001	0.006
	scalar	282	0.951	0.050	0.045	−0.011	0.005	0.004
	scalar_part_	281	0.957	0.047	0.044	−0.005 ^a^	0.002 ^a^	0.002 ^a^
	strict	299	0.957	0.045	0.045	0	−0.001	0.001
Age	configural	508	0.962	0.047	0.041			
	metric	562	0.959	0.046	0.052	−0.003	−0.001	0.011
	metric_part_	559	0.961	0.045	0.049	−0.001 ^b^	−0.001 ^b^	0.008 ^b^
	scalar	589	0.959	0.045	0.050	−0.002	0.000	0.001
	strict	643	0.951	0.047	0.054	−0.007	0.002	0.003

Note: df = degrees of freedom, CFI = comparative fit index, RMSEA = root mean square error of approximation, SRMR = standardized root mean square residual, ΔCFI = difference in CFI, ΔRMSEA = difference in RMSEA, ΔSRMR = difference in SRMR. ^a^ Difference in comparison to the metric model. ^b^ Difference in comparison to the configural model.

## Data Availability

Restrictions apply to the availability of these data. The data are available from the authors with the permission of the PI (Alexander Grob) of the original standardization study.

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
