# Peer review of "Investigating the Relation of Intelligence and Executive Functions in Children and Adolescents with and without Intellectual Disabilities"

_children, 2022, doi:10.3390/children9060818_

Round 1

Reviewer 1 Report

This paper is written very clearly and the statistical methods was valid. I only had one minor comment on the paper:

1)  I would suggest on include M1 model fit results in Table 1 so that readers can compare M1 versus other models' fit results.

Author Response

We would like to thank you very much for your review. We intended to focus on the main results and therefore highlighted the second orders models in Table 1. However, we do see the advantage of better comparability and have included M1 in the table now.

Reviewer 2 Report

In this manuscript, the authors examined the factor structure of executive function (EF) and how it relates to a well-established intelligence factor. They also considered whether the structures of EF and intelligence differ by age, sex, and borderline intellectual disability. The manuscript is well-written, and the methodology and analytic approach are strong. It stands to make an important contribution to the literature on EF and intelligence. See below for a few comments for the authors to consider:

Introduction

  • On page 2, the authors note that theories of intelligence are tightly coupled with measurement. However, there is no discussion in the introduction about how intelligence is actually measured and, more specifically, how the IDS-2 is similar (or different) from other measures of intelligence (for which there are many; e.g., Stanford-Binet, Wechsler, etc.). This would provide important context for interpreting the findings with regards to the IDS-2.
  • Given that there is such a large age range of the participants, and that a main aim of the study is to examine measurement invariance by age, the authors might consider discussing how the relation of intelligence and EF might change with age (if space allows). They do a nice job describing how EF changes from early childhood to early adulthood, and it would be helpful to have a similar discussion in section 1.2.

Method

  • It is not clear to me why some participants would have completed 14 subtests and others only completed 7 subtests (pg. 6, lines 253-254). Did the testing procedure differ across time or perhaps there are participant characteristics that dictate how many subtests they complete?

Results

  • I am wondering whether the authors had considered conducting a separate EFA with just the subsample of participants with borderline intellectual disability rather than a multigroup analysis to assess measurement invariance. It might be helpful for researchers to have some idea of the factor loadings and model fit to guide future work.

Discussion

  • Introducing the issue of cross-loadings and model fit for the borderline intellectual disabilities analyses felt a bit out of place in the discussion. Rather, it seems like it would make the most sense to describe these issues in the results section along with the accompanying statistics.

Other notes:

  • There were some minor errors throughout. For example:
    • pg. 3, line 119 should be “working memory”
    • pg. 5 line 217 should be “this research question” rather than plural
    • pg. 7 line 295 should be “the latter” not the ladder
    • pg. 11 line 511 appears to be missing a word: “was largely … across sex and age group”

Author Response

We would like to thank you for your constructive and detailed input on our current research paper. Below we have outlined our corrections or explanations in accordance to the chronology of your recommendations.

We addressed the first recommendation regarding the comparability of the IDS-2 to other intelligence measures by adding references to the Wechsler Intelligence Scale for Children – Fifth Edition and the Stanford-Binet Intelligence Scales - Fifth Edition to point out their resemblance regarding their hierarchical structure and the difference in the number of the included broad abilities and hope to thereby provide appropriate context for the reader. However, we consider that a more in depth discussion of the similarities and differences of different intelligence measures might lay behind the scope of the current article, as we cannot provide an empirical comparison of these instruments.

Regarding the development of the relation of intelligence and executive functioning, we tried to incorporate some information within section 1.2 and hope that this section better supports our third research aim now.

We would like to thank you for pointing the lack of clarity regarding the number of subtests. The IQ profile can only be calculated when all 14 subtests are administered. Therefore, we used the IQ score based on 7 subtests for subsample division, when there were missing values in some subtests preventing the calculation of the IQ profile. We realized no systematic drop-out effects like age, ability level or gender. We added a more detailed description and hope that the subsampling procedure is now more clearly stated within the article.

To address your input on possible exploratory analyses and the discussion of cross-loadings, we added the results on the model without a cross-loading and some additional exploratory analyses based on the borderline intellectual ability subsample within our results section and discussion.

We corrected all minor errors according to your annotations.

Reviewer 3 Report

Thank you for the opportunity to review this manuscript. The current msc is about a topic of relevance and general interest to the readers of the journal (and the world). I have few comments. The introduction is clear, comprehensive and well documented and updated, and provided a solid rationale for the study; reflects very recent contributions on the subject of study. The aims of the research are also clearly exposed. The objective is well-founded, the methodology seems adequate and well implemented, although I question below some issues in this regard.

The authors should clarify the title (or the text), because it mentioned children/adolescents with and without disability and the sample comprises participants with IQ<85, which is not aligned with the most recent definition of Intellectual and Developmental Disability (IDD, two standard deviations below the mean). Further, was the participants’ adaptive behavior analyzed? Because at page 5, lines 197-198 authors state that “diagnosing and intellectual disability requires an individually administrated, standardized psychometric test […] resulting in an intelligence quotient (IQ) of 70 or under”. And this is not totally correct. So, how was the participants with IDD selected (inclusion criteria)? It would be helpful to know how individuals were differentiated with regards to falling in either the borderline, mild or moderate. The data was collected in 2017 – how can be assured that it still is current? I do not completely understood why some variables (e.g., residency, maternal education status) were analyzed and how these results were considered within discussion. How was the groups of IDD’ sample size calculated? Please justify the sample size adequacy (NIDD=255 – maybe authors could add a brief explanation of the adequacy of this number). What is the worth-value of the instrument chosen compared to others? One of the major critics of the assessment of executive functions is the fact of being laboratorial and do not have an ecological approach/impact. I understand the tension between space and content, but it would be good to provide some brief information on (a) psychometric properties and b) systems of collecting data by the original authors. Given the very detailed results, I would appreciate the main findings being drawn out in the Discussion. What are the key messages the reader should take from this study? It needs to go beyond the descriptive and answer the 'so what?' question - i.e., why are these results important and how can they be used? One of the recommendations is that author(s) are invited to strengthen the discussion about findings: what is the worth-value of such a study and how findings could be “used” on others countries? Discussion could also be strengthened by providing detailed interpretation of data and about the significance of the findings of the work. Somehow, I feel that it lacks a conclusion: usually, the finding highlighted here relates to the primary outcome measure; however, other important or unexpected findings should also be mentioned. It is also customary, but not essential, for the authors to express an opinion about the theoretical or practical implications of the findings, or the importance of their findings for the field. Thus, the conclusions may contain three elements: The primary take-home message; The additional findings of importance; The perspective.

Author Response

We would like to thank you for your input on our current research paper. Below we have outlined our corrections or explanations in accordance to the chronology of your recommendations.

Thanks a lot for this critical and constructive recommendation regarding the operationalization of intellectual disabilities. We changed the title into: Investigating the Relation of Intelligence and Executive Functions in Children and Adolescents with and without BODERLINE Intellectual Disabilities. We hope this clarifies that we referred to a broader range of cognitive functioning than suggested by the original title.

We conducted the study based on a research conceptualization of intellectual disabilities that is based on cognitive functioning as there was no complete diagnostic included in the standardization process. We are aware that the clinical diagnosis of intellectual disability requires additional information and encourage future research including a more comprehensive clinical assessment. Furthermore, a differentiation of the degree of intellectual disability was not possible within the current study due to limited sample size. We hope that our changes in the method section and discussion of our manuscript clarify our approach.

Regarding the current relevance of our study, we would like to argue that the analyses within our study were based on the most recent German standardization sample of the IDS-2. Therefore, current assessments with the IDS-2 in practical settings is therefore are based on the very same sample and hence we consider our highly relevant as being relevant for current practical applications. However, we added the need to revisit the structural relations under investigation in different (and possibly more current) samples to our discussion.

Thank you for pointing out, that we did not refer to our control analyses regarding the socio-demographic variables within our discussion. We now refer to this results and shortly address possible confounding effects. Moreover, we included possible implications of the sample size within our discussion.

We consider the main worth-value of the IDS-2 the joint assessment of cognitive abilities, including executive functions and other areas of childhood development within one instrument. This enables a better comparability as interpretation are based on the same standardization sample. We hope that this is now more strongly incorporated in our manuscript.

With regard to your recommendation to include (a) psychometric properties and b) systems of collecting data by the original authors we consider that this information is already partly included in the method section. Because of the focus of the paper and the length, we consider a more detailed description to be unnecessary.

Thank you for highlighting your perspective on the weaknesses of our discussion and the missing conclusion. We hope to have sufficiently addressed these issues by our changes and think that these changes strengthened our discussion.

Moreover, we included a conclusion to clarify our focus, results and limitations and hope to have thereby improved the paper.

Reviewer 4 Report

I think it is a good article, I would only update the bibliography.

Author Response

Thank you very much for your review. We checked the bibliography for errors and updates.

Reviewer 5 Report

Current manuscript investigated structural relations between intelligence and executive functioning (EF), and tested an integrative model for measurement invariance regarding sex,  age, and intellectual disability. It showed structural relatedness between intelligence and EF, as well as measurement invariance for sex and age groups. However, invariance of the integrative model for the intellectual disability subsample could not be established due to the model determination problems. The paper is well written, analyses are justified and described in detail, conclusions closely follow the obtained results. Below are some comments that have to be addressed before the publication of the manuscript. I am also concerned that this manuscript would be a better fit for a scientific journal on intelligence rather than the Children journal since it does not cover developmental trends in children, but instead looks at the structure of intelligence and EF in 8-20-year-olds.

Major comments:

line 267: “The parents (for 5- to 15-year-olds)” – researchers have previously specified that participants younger than 8 years old were not included in the current study…

I noticed some inconsistency between the Results and Discussion: on line 488, the authors reported that “boys and girls seem to differ regarding their mean performance in the GS subtest” (mean across all ages or mean in each of the four age groups?), while on line 561, they stated intercept differences: “the intercept of the GS subtest measuring visual-spatial abilities differed between sexes, with boys scoring 0.31 SD higher than girls” (differences at age 0 years old?). The report should be consistent between the Results and Discussion sections.

Minor comments:
line 178: “highly heritably” – should be “heritable.”
line 186: “Accordingly, we assume that the EF domain can be integrated an additional broad ability into ...” – should be “integrated as an additional…”
line 222: “and it can be assumed that a common EF factor might comparable to other primary factors of intelligence” – should be “might be comparable…”
line 295: “the former refers to auditory information, the ladder to visuo-spatial information” – should be “the latter…”
line 377: “its fit within our sample was re-assed utilizing a confirmatory factor analysis…” – reassessed?
line 600: “part of our study describe beneath” – should be “described beneath”
line 610: “results can be assumed to applicable” – should be “to be applicable”

Author Response

We would like to thank you for your detailed input on our current research paper. Below we have outlined our corrections or explanations in accordance to the chronology of your recommendations.

With regard to your overall comment we would like to argue, that the IDS-2 is a current instrument that is available in twelve different languages and is widely used in clinical and educational diagnostics in German-speaking countries and increasingly in other countries. Therefore, we consider the results on the structure and relation of the included cognitive abilities to be highly relevant for both research and practical applications. In addition, the current Special Issue is dedicated to “The Research of Intelligence and Specific Developmental Disorders in Children”. Therefore, we think our paper fits the scope of the journal and more specifically the current special issue well.

We corrected the inconsistency regarding the age range of participants by changing it in accordance to our focus in the current paper.

We decided to exclude the reported mean comparison, as the interpretability of intercept differences is limited and the inclusion of mean comparisons possibly leads to misinterpretations.

We corrected all minor errors according to your annotations.

Round 2

Reviewer 3 Report

The authors attempted to provide  very constructive modifications to their work based on the feedback they received that is much appreciated.

Author Response

We really appreciated the constructive feedback you have given us and would like to thank you for that!